# Mitochondrial Connexins and Mitochondrial Contact Sites with Gap Junction Structure

**DOI:** 10.3390/ijms24109036

**Published:** 2023-05-20

**Authors:** Selma Cetin-Ferra, Sharon C. Francis, Anthonya T. Cooper, Kit Neikirk, Andrea G. Marshall, Antentor Hinton, Sandra A. Murray

**Affiliations:** 1Department of Cell Biology, University of Pittsburgh School of Medicine, Pittsburgh, PA 15261, USA; scf28@pitt.edu (S.C.-F.);; 2Department of Physiology, Morehouse School of Medicine, Atlanta, GA 30310, USA; sfrancis@msm.edu; 3Department of Molecular Physiology and Biophysics, Vanderbilt University, Nashville, TN 37232, USA; 4Department of Biology, University of Hawaii, Hilo, HI 96720, USA

**Keywords:** mitochondria, connexin, annular gap junctions, contact sites

## Abstract

Mitochondria contain connexins, a family of proteins that is known to form gap junction channels. Connexins are synthesized in the endoplasmic reticulum and oligomerized in the Golgi to form hemichannels. Hemichannels from adjacent cells dock with one another to form gap junction channels that aggregate into plaques and allow cell–cell communication. Cell–cell communication was once thought to be the only function of connexins and their gap junction channels. In the mitochondria, however, connexins have been identified as monomers and assembled into hemichannels, thus questioning their role solely as cell–cell communication channels. Accordingly, mitochondrial connexins have been suggested to play critical roles in the regulation of mitochondrial functions, including potassium fluxes and respiration. However, while much is known about plasma membrane gap junction channel connexins, the presence and function of mitochondrial connexins remain poorly understood. In this review, the presence and role of mitochondrial connexins and mitochondrial/connexin-containing structure contact sites will be discussed. An understanding of the significance of mitochondrial connexins and their connexin contact sites is essential to our knowledge of connexins’ functions in normal and pathological conditions, and this information may aid in the development of therapeutic interventions in diseases linked to mitochondria.

## 1. Introduction

It is becoming apparent that when investigating mitochondria, it is not enough to consider them as isolated organelles, but instead, there is a need to study their contacts and communication with other organelles that either regulate their function by sharing molecules or are impacted by their mitochondrial association. While contact sites have been demonstrated to occur between most organelles [1], including mitochondria, contact sites between mitochondria and organelles composed of gap junction proteins has received little consideration. However, the finding of members of the gap junction family of proteins, connexins, on mitochondrial membranes [2,3,4,5] and the reports of mitochondrial contacts with gap junction plaques at the plasma membrane and vesicles composed of connexins in the cytoplasm [6,7,8,9] has opened a new realm of possible functions for connexins, outside of their primary cell–cell communication functions. We will discuss connexins in general and then go on to present what is known about mitochondrial contacts with connexin-containing membranes and mitochondrial connexins.

Connexins were first identified in 1986 as a family of membrane proteins that formed the gap junction channels on the plasma membrane [10]. There are twenty-one members of the connexin family in humans [11,12] that differ by their amino acid sequences and molecular weights [10]. These various connexins are differentially expressed throughout the body in a cell type-specific manner, with connexin 43 (Cx43) being the most ubiquitously expressed [13,14]. During channel formation connexins assemble into six-unit complexes, namely hemichannels, which are transported to the plasma membrane in secretory vesicles [12,15,16]. Once on the plasma membrane, the hemichannels form two contacting cells dock with one another to form gap junction channels that then cluster into the gap junction plaques. The channels in these gap junction plaques open and close to regulate the movement of molecules, which are 1000 Da or less, between cells [11,17]. Additionally, beyond size, shape and charge of the transported molecules, permeability is influenced by the size of the open channel, the connexin isoform types to form the channel, the channel pore lining and non-pore lining connexin residues, which contribute to the pore shape [18,19,20]. The cell–cell communication of molecules is dependent on the presence of open pores. The removal of gap junction channels from the cell surface decreases the number of channels available to participate in cell–cell communication. One method for removing “old” gap junction channels from the plasma membrane involves an internalization process in which a portion or the entire gap junction plaque is taken into one of two contacting cells [21,22,23,24,25,26,27,28,29] (Figure 1). We and others have demonstrated that this internalization process is mediated by clathrin and its adaptor proteins that interact with the channel connexins, and result in the invagination of gap junction plaques to form gap junction buds [22,30,31]. Subsequently, dynamin then associates with the neck of these buds to separate them from the plasma membrane, thus resulting in the release of annular gap junction vesicles into the cytoplasm of one of the two adjacent cells [21,22,23,24,25,26] (Figure 1). The “gap” seen between the double membranes of the annular gap junction vesicles is the same width as that measured between the membranes of the gap junction plaque membranes. The annular gap junction is thus a double membrane vesicle composed of gap junction channels that were derived from hemichannels from the donor and host cells (Figure 1). While the exact functions of these structures need to be further elucidated, annular gap junctions, known as connexosomes, are thought to play a role in the turnover of old or damaged gap junction channels via their removal from donor and host cells for either lysosomal degradation or the reformation of functional gap junctions [6]. While, to our knowledge, there are no reports of connexin secretory vesicle membranes forming contacts sites with mitochondria, mitochondria have been reported to form contact sites with both annular gap junction vesicles and gap junction plaque membranes [6,9] and connexin proteins have been reported on the outer and inner mitochondrial membranes that suggestive of connexins playing a role in mitochondria activity [32,33]. The findings of connexins within the mitochondria and the reports of mitochondrial contacts with annular gap junction vesicle and gap junction plaque membranes were once met with skepticism. It has been suggested that the mitochondrial outer membrane may be permissive for protein insertion and thus non-mitochondrial proteins could potentially be mistargeted to the mitochondria that lack function [34]. Further, it was suggested that biochemical mitochondrial isolation methods resulted in contaminations with proteins and other membrane organelles. However, based on the observations made with a combination of different imaging techniques and the development of new tools for visualizing, localizing and tracking connexins, it has become evident that contrary to the dogma that connexins only participate in cell–cell communication between cells, they are found on mitochondrial membrane and thus may have other functions in the cell unrelated to cell–cell communication [4]. Furthermore, connexin hemichannels have been found on the plasma membrane, where it is suggested that they function in cell injury and cell death, and on the nuclear membrane, where they are thought to play roles in modulating gene expression [35,36]. Such observations introduce new concepts and possibilities for connexin non-canonical functions and represented a major shift in how we once thought about connexin function.

Connexins have been demonstrated in the mitochondria of a number of different cell types that include coronary endothelial [37], liver [3,38,39], astrocytes [40] and retinal endothelial cells [41]. In the heart, connexins have been shown to be localized to the inner mitochondrial membrane of cardiomyocyte [38,42], where it is speculated that they play a role in cardioprotective signaling [43,44,45,46,47]. What we know about the function and presence of connexins in the mitochondria is still emerging and is poorly understood. The intent of this review is to highlight the information on mitochondrial interactions with gap junction plaques and annular gap junction vesicles and to present information on mitochondrial Cx43 translocation, location and function. Further, relevant investigation into mitochondrial connexins could improve the understanding of mitochondria and connexin function in cell behavior, as well as open new avenues for controlling normal and pathological cell physiology.

### 1.1. Mitochondrial/Gap Junction Plaque Contact Sites

Mitochondrial interactions with gap junction plaque membranes have been demonstrated in vertebrate and invertebrate cells with transmission electron microscopy [7,8,48,49], reviewed by Montes de Oca Balderas [50] and by Peracchia [9]. In some transmission electron microscopic images, bridge-like structures that appear to align with or tether the mitochondria to the gap junction plaque membrane reported in crayfish lateral giant axons were suggested to be consistent with the formation of contact sites between mitochondrial and gap junction plaque plasma membranes [9,49]. The molecular makeup of the electron-opaque particles that form bridge-like structures between mitochondria and gap junction plaques is yet to be identified. Based on the similarity of electron-opaque particles in size and spacing to gap junction particles seen in thin section micrographs from tangentially cut crayfish mitochondria [51], it has been speculated that gap junction channels at the plasma membrane may be capable of binding with the particles on the mitochondrial membrane. The implication was that the electron-opaque particles were mitochondrial gap junction hemichannels, composed of gap junction proteins.

In further support for the existence of mitochondrial/gap junction contact sites is the finding of the closeness of the space of separating between the gap junction plaques and the mitochondrial membranes at the contact sites. The approximate spacing at the contact sites was shown to be 20 nm or less, as determined from measurements obtained with transmission electron microscopic techniques from the micrographs of mammalian myocardial cells [49]. This spacing is similar to that observed between mitochondria and the endoplasmic reticulum (of 10–30 nm) [52,53] and a spacing which has been shown to allow the endoplasmic reticulum to regulate mitochondrial fissions [52]. In addition, gap junction plaque membrane profiles were often shown to conform to the shape of the contacting mitochondria (Figure 2 and Figure 3), a pattern suggestive of a functional significance, as opposed to representing the random association of these organelles in a highly packed area [6,49,54]. Since it takes energy for an organelle to conform to the contour of another organelle, a process not dependent on random events has been suggested, which is consistent with a functional contact [1]. A close association rather than random association has also been suggested, based on the presence the bridge-like strands of material in the space between the mitochondria and gap junction plaques [49]. In some cases, mitochondria from two different cells were demonstrated to line up adjacent to either side of the gap junction plaque membrane rather than being randomly distributed near the gap junction plaque membrane [49]. Remarkably, in mice, over 40% of the length of gap junction plaque in the ventricular wall was observed to be closely associated with mitochondria [49]. It is also possible that this close association between mitochondria and gap junctions reflects a functional relationship, such as the provision of ATP to support the energy demands of intercellular communication mediated by gap junctions; however, whether these associations have a functional role needs to be elucidated. The observation in the cardiomyocytes of mitochondria near the plasma membrane rather than randomly assorted throughout the cell has long been thought to reflect the positioning of the mitochondria relative to where the oxygen is entering the cell and where energy demands are high. It could be speculated, given the close association of mitochondria with gap junction plaques in the ventricle, that there is a direct relationship been the energy demanding processes and mitochondrial location. This speculation needs to be experimentally tested. However, in genetic models where Cx43 in heart was either replaced with connexin 32 (Cx32) or ablated, a reduction in mitochondrial membrane potential and alteration in mitochondrial respiration and oxygen consumption were observed [55,56,57]. This suggests that mitochondrial Cx43 is related to mitochondrial function, which includes energy demand. A putative proposed model based on ultrastructural observations is that the gap junction channels on the plasma membrane dock with the hemichannels on the outer mitochondrial membrane (See Section 1.3). While this is an interesting hypothesis, experimental data supporting this model are needed.

The detailed descriptions of contact sites between mitochondria and gap junction plaques were provided through freeze-fracture electron microscopic observations [9,58,59,60,61]. With this powerful method, the intimate relationship between gap junction and myocardial mitochondrial membranes can be appreciated. In fact, the two membranes can be observed to be superimposed, again pointing to a close association of these two organelles [9,49]. Further with lanthanum nitrate tracer techniques, the spaces between what appears to be the mitochondrial membrane and a gap junction plaque membrane were delineated via the electron dense lanthanum tracer [7]. While the typical hexagonal morphology of gap junction plaque membrane was definitively seen with this technique [61], it was difficult to positively discern mitochondria membranes in these lanthanum tracer studies. Furthermore, with this technique, it was difficult to visualize how the membranes of the two organelles might be tethering to one another, as suggested by observations made with transmission electron microscopy. However, lanthanum nitrate techniques do allow invaginated surface gap junction plaque membranes to be distinguished from annular gap junction vesicles, which are detached from the cell surface and thus free of lanthanum tracer. It should be noted that while there is morphological evidence consistent with mitochondrial/gap junction plaque contact sites, biochemical studies are needed to support these morphological observations. Further, it was speculated that the mitochondria/gap junction plaque contact areas may serve to buffer the intracellular Ca^2+^ concentration within the area of the gap junction plaques and thus regulate the ionic permeability of the gap junction channels. This possibility was suggested based only on the known role of mitochondria in sequestering Ca^2+^ ions and not on actual experimental data. The benefit of the close association of the mitochondria and gap junction plaques, the interactions and the identities of the molecules located at the contact sites need further investigation. Of equal need for consideration are the presence and role of contacts between mitochondrial and annular gap junction vesicle membranes. 

### 1.2. Mitochondrial/Annular Gap Junction Vesicle Contact Sites

Before the time-lapse imaging of tagged connexin and super-resolution microscopy was readily available, it was difficult to definitively distinguish annular gap junctions from an invaginated annular profile of gap junction plaques cut at an angle with transmission electron microscopic techniques, as they appeared annular. Thus, while there are reports of mitochondria in contact with invaginated gap junction plaque membranes [49,62], the information on contacts with annular gap junction vesicles remains limited. In the early literature, it was rarely suggested that annular gap junctions actually existed. In two different early studies, for example, the authors speculated that convoluted or invaginated surface gap junction plaque profiles cut at an angle could be observed in the cytoplasm that made them what appeared to be annular vesicles [49,62]. However, with the development of new and combined imaging techniques, annular gap junction formation could be visualized with time-lapse microscopy and the complete separation of the invaginated gap junction plaque membrane from the plasma membrane was demonstrated [6,21,24,26,63]. The size of the annular gap junction released into the cytoplasm was shown to reflect the size of the gap junction plaque membrane area being internalized. Furthermore, with confocal microscopy and computer-assisted 3D reconstruction methods, annular gap junction vesicles could be definitively demonstrated to be free within the cytoplasm and not invaginated gap junction plaque membrane, as suggested in early studies. Further, it has been shown that the majority of annular profiles seen in the cytoplasm with transmission electron microscopy are annular gap junctions [6].

Annular gap junction/mitochondria contacts have been demonstrated with immunocytochemical, super-resolution imaging combined with 3D-reconstruction, as well as transmission electron microscopic techniques (Figure 2 and Figure 3) [6]. With immunocytochemical methods coupled with super-resolution microscopy, the central lumen of the annular gap junction was revealed, which helped to positively identify and distinguish these structures from other possible cytoplasmic structures. In addition, immunocytochemical 3D rotations results confirmed that the Cx43 puncta contacting the mitochondria were annular gap junction vesicles and not convoluted surface membranes cut at an angle to appear annular, as suggested in earlier reports [49,62]. A problem with immunocytochemical colocalization was the difficulty in determining the space between the two organelles at contact sites and the possible interpretation of contact when two organelles were only close to one another.

With transmission electron microscopy, in some cases, mitochondria and annular gap junction vesicles appeared to be in direct physical contact with one another, suggestive of a biochemical active contact site between the two organelles [6] (Figure 2 and Figure 3). As described for contact sites between mitochondria and gap junction plaques, mitochondria contours conformed to the shape of annular gap junction vesicles at contact sites. The average distances between mitochondria and annular gap junctions, however, measured using ultrastructural imaging techniques at contact sites in cells of adrenal origin in cell cultures, were on average 18.4 ± 0.7 nm, ranging from 0–25 nm [6]. It was noted that the mitochondria/annular gap junction vesicles contact distance was comparable to the distance measured between the mitochondria and the endoplasmic reticulum (of 10–30 nm) [53], gap junction plasma membranes (10–20 nm) [9,49], and the lysosomes (9.57 ± 0.76 nm) [64] at their contact points. The function of endoplasmic reticulum–mitochondria contact sites has been well characterized [54,58,65,66] and shown to function in mitochondrial fission [53,64,67,68]; however, the role of mitochondrial contacts with annular gap junction vesicles can only be speculated upon. The possible delivery of used connexins from the plasma membrane to mitochondria is one possibility. Furthermore, mitochondria have been observed via transmission electron microscopy within the interior of large annular gap junction vesicles [69,70,71]. It has been suggested that mitochondria located in the cytoplasm near invaginating gap junction plaques were trapped within annular gap junction vesicle lumen during their formation. It is thus thought that annular gap junctions may facilitate the transfer of mitochondria as well other cytoplasmic components between cells [67,68,69,70,71]. This introduces a plethora of possibilities for cell regulation by the transferring materials using annular gap junction vesicles and other vesicles engineered to express connexins. Similarly, the transfer of material from exosomes may be facilitated by the connexin-mediated docking and speculated internalization into a cell [72]. Exosomes have been demonstrated to form connexin hemichannels that facilitate the transfer of their exosomal cargo to targeted cells [73]. Additional information is needed regarding the role of the connexin-mediated transfer of material via annular gap junctions and exosomes. Such information would impact the development and the design of innovative therapeutic methods for treating defective cell populations via connexin-mediated delivery of the materials, perhaps even healthy mitochondria.

Although putative theories have been presented to explain the purpose of the mitochondria gap junction containing structure contacts, data are needed to answer important questions on the molecules and mechanisms involved in establishing contact sites and identifying the cellular benefits of such sites. Insight into the role of these contact sites will be enhanced by obtaining information on the orientation of connexins at contact sites. Specifically, the greater understanding of the possible proteins and structural entities that may serve to tether the mitochondria to connexin-containing structures may elucidated if connexin can pass from the gap junction membrane to the mitochondrial membrane. Studies are needed that explore the possibility of direct mitochondrial communication with either adjacent cells (in the case of the gap junction plaque contact sites) or with the contents of the annular gap junction vesicle lumen. Theories and explanations of how mitochondria and gap junctions containing structures may interact have resulted in a several interesting hypotheses and speculations on the interactions of mitochondria and connexin-containing membranes, which are yet to be verified. It is thought that understanding and visualizing the physical nature of the mitochondrial/annular gap junction membrane interaction could shed light on the functions of these contact sites.

### 1.3. The Mechanism of Mitochondria/Gap Junction Structure Interactions

Researchers have suggested, based on transmission electron and freeze-fracture microscopy, that connexins found in hemichannels on the mitochondrial outer membrane may bind with gap junction plaque or vesicle channels [9,61]. The theories of the direct interaction of the gap junction vesicles and plaques channels with mitochondria are based on transmission electron microscopic observations of the bridge-like structures seen between gap junction and mitochondria and existing knowledge about the connexin molecular structure [74,75,76]. Connexins are integral transmembrane proteins with an N-terminal region, cytoplasmic loops, two extracellular loops that project from the plasma membrane surface into the extracellular space and a carboxyl-terminal tail segment (Figure 4A). When gap junction proteins that form hemichannels at the plasma membrane dock with one another to form the gap junction channel, they do so via interactions that occur in their extracellular loops, while the N-terminal region and carboxyl terminal tail of the hemichannel are within the cytoplasm [75] (Figure 4A,B).

The suggestions of direct or indirect interactions between connexins found on the mitochondrial and gap junction structures at contact sites are highly hypothetical. There is a need for future studies designed to determine whether the bridge-like structures seen through transmission electron microscopy on mitochondria are connexins, demonstrate the presence and orientation of connexins in the outer mitochondrial membrane and identify molecules at contact sites between mitochondria and gap junction plaque and annular gap junction vesicle membranes.

Elucidating the identity of the material observed in the space at contacts between the mitochondria and gap junction plaque is needed. Particularly, in some studies this material looks too diffuse and projects too far into the cytoplasm to be explained by docking between the hemichannels, carboxyl-terminal to carboxyl-terminal or extracellular loop of the gap junction and outer mitochondrial membranes. An explanation could be that the interactions at mitochondrial/gap junction plaque membrane contact sites are indirect and involve intervening molecules. Although proteins between the mitochondria and plasma membrane sites [82] and between mitochondria and endoplasmic reticulum contact sites [83] have been identified, to our knowledge, no such information is available for molecular interaction at mitochondrial and connexin gap junction structure contact sites. However, using proteomic techniques, connexins have been demonstrated to interact with mitochondrial proteins, the superoxide dismutase 2 and ATP synthase ATP5J2 subunit, in non-pathological states [84]. There is a critical need to investigate mitochondrial/connexin structure contact site and the possible molecules involved in tethering the two organelles. In indirect interactions, either the carboxyl-terminal tail of the gap junction plaque membrane or the intracellular loop could bind other structures and thus tether the mitochondrial and gap junctional membranes. While such speculations are interesting, it should be noted that the presence of hemichannels in the outer mitochondrial membrane has only been provided by the limited observations of particles on mitochondria seen in tangential cuts through the organelle. While it has been suggested that such particles line up at the contact site to allow for material to transfer between these two organelles [3,9,48,60], there are no experiments that support this hypothesis. Furthermore, the presence of connexins on the outer membrane is very controversial and it is certainly not clear whether they form the electron opaque structures seen in transmission electron microscopic observations. The mitochondria–ER cortex anchor (MECA) complex and siderioflexin-1 (SFXN-1) could also be located at mitochondrial/gap junction membrane contact sites, given their known roles in binding the mitochondria to the plasma membrane and the endoplasmic reticulum [85]. Based on the intimate physical association of the membranes of the mitochondria and the annular gap junction, connexin has been postulated to be delivered to mitochondria at contact sites between the two organelles [69]. Notably, pannexins, another family of gap junction proteins that are expressed mainly in invertebrates [86], have been shown at mitochondrial/ER contact sites [87]. Furthermore, early observations of the bridge-like particles at mitochondrial/gap junction plaque membrane sites were made in cray fish [48,60]. In this invertebrate system, the bridge-like particles would be composed of pannexins [48,60]. It is possible that pannexins may be present on mitochondrial membranes, where they have specific functions or functions attributed to connexins. At mitochondrial/ER contact sites, pannexin 2 has been speculated to regulate apoptosis [88]. Studies identifying pannexin’s and connexin’s cellular locations, functions and their specific roles at contact sites are needed to increase the understanding of gap junction/organelle interactions. Regardless of the controversy surrounding the role of gap junction/mitochondrial contact sites, it has become clear that mitochondria contain connexins [3,38,39,89,90].

### 1.4. Presence of Connexins within Mitochondria

Several members of the connexin family have been reported in mitochondrial membranes, which include connexin 40 (Cx40) in coronary endothelial cells [37], Cx32 and connexin 26 (Cx26) in liver mitochondria [3,38,39,89,91] and Cx43 in a wide variety of different cells [38,55]. Proof of the presence of connexin in mitochondria has been provided by researchers who used a combination of imaging and molecular biological approaches. The imaging techniques used were light and super-resolution immunocytochemistry and immunogold transmission electron microscopy [33]. Flow cytometry, Western blot analysis [33,38,91] and subcellular fractionation combined with immunoprecipitation-high throughput mass spectrometry and reciprocal co-immunoprecipitation (co-IP) [38,92] were molecular approaches used to provide proof for the presence of connexin within mitochondria. The most evidence for connexin in the mitochondria, however, comes from the studies of Cx43, the most widely expressed member of the connexin family [93]. Mitochondrial Cx43 was reported to be differentially expressed in the heart, such that it is not found in cardiomyocyte interfibrillar mitochondria but is present on the inner membrane folds (cristae) of cardiomyocyte subsarcolemmal mitochondria [94,95]. Some studies have now confirmed connexin localization to the inner membrane cristae [42,55,89], which coincides with the location where the oxidative phosphorylation machinery and various transporters are housed and where most of the essential mitochondrial functions occur. Cx43 protein was further isolated from mitochondria in cardiomyocytes specifically concomitant with ischemic preconditioning. Connexin 43 has been reported on the outer mitochondrial membrane of heart cells [96], while the presence of Cx32 has been shown on the outer mitochondrial membrane of certain cells in the liver [3].

Within the subsarcolemmal mitochondria, Cx43 delivery to the mitochondria requires heat shock protein 90 (Hsp90) and the translocase of the outer membrane 20 (TOM 20) [39,91]. While the orientation of the Cx43 in the outer mitochondrial membrane has not been demonstrated, the orientation of the Cx43 carboxyl-terminal tail and extracellular loop relative to the mitochondrial matrix has been demonstrated. In elaborate studies, digitonin or triton were used to rupture the outer mitochondrial membrane, and thus expose the inner mitochondrial membrane, and Cx43 antibodies were used to detect either changes to the carboxyl-terminal tail or the “extracellular loops” of mitochondrial connexins found on the inner membrane [38]. In these preparations, it was demonstrated that Cx43, once within the inner mitochondrial membrane, is positioned with its carboxyl-terminal tail within the intermembrane space. The Cx43 extracellular loop regions would be in the mitochondrial matrix compartment (Figure 5). This Cx43 orientation was further suggested in studies in which different amounts of proteinase K were used to remove the outer mitochondrial membrane while leaving the inner membrane with its integral proteins, including Cx43, intact. The carboxyl-terminal tail, but not the extracellular loops, were altered by this procedure, thus consistent with the Cx43 hemichannel extracellular loops being within the matrix rather than within the mitochondrial intermembrane space. It was suggested that in this orientation, Cx43 hemichannels would be open, since the extracellular (“matrix loops”) would be exposed to a negative mitochondrial membrane potential of −150 to −180 mV, while the carboxyl-terminal tail region of Cx43 would be exposed to a mitochondrial membrane potential of −70 to −80 mV [97]. The net difference (+70 to +110 mV) between the matrix loops and the carboxyl-terminal tail would be above the threshold for the voltage-dependent opening of the Cx43 hemichannels (+40 to +50 mV) in ventricular cardiomyocytes [97,98]. Studies are needed to confirm whether the mitochondrial Cx43 hemichannels are open and functional. However, if the mitochondrial hemichannels are continuously open, one would question how this could be a benefit rather than a detriment to the cell.

### 1.5. Function of Mitochondrial Cx43

The reported findings of a relationship between the level of Cx43 in the mitochondria and mitochondrial function provide evidence that Cx43 is playing a functional role in mitochondria and cell physiology [45,56,57]. Specifically, alterations in mitochondria Cx43 levels have been demonstrated to result in changes in mitochondrial metabolic activity and mitochondrial morphology, consistent with a functional mitochondrial Cx43 dependency. The most information on the functional role of mitochondrial Cx43 comes from studies of the heart where mitochondrial Cx43 has been suggested to regulate potassium flux and mitochondrial respiration [38,55,57,99].

Potassium flux, mitochondrial respiration and reactive oxygen species (ROS) production have been shown to be regulated by Cx43 in cardiomyocyte subsarcolemmal mitochondria [38,55,57,99]. Specifically, Boengler and colleagues demonstrated that the treatment of isolated cardiomyocyte subsarcolemmal mitochondria with an inhibitor of Cx43 hemichannels, Gap19, caused a ~40% decrease in the inward potassium influx [99]. Potassium flux was similarly depressed in Cx43-deficient isolated cardiomyocyte subsarcolemmal mitochondria. In contrast, Gap19 did not affect potassium flux in isolated cardiomyocyte interfibrillar mitochondria, which are devoid of Cx43. This pharmacological and genetic evidence is consistent with mitochondrial Cx43, playing a role in mitochondrial potassium influx in some cells [55,57]. There is a need to determine why some cells have mitochondrial Cx43, while others in the same tissue do not. Understanding the needs of certain cell types to traffic Cx43 to the mitochondria will increase our understanding of Cx43 trafficking and provide information on decoding mitochondrial functions.

In adipocyte cell cultures, the down-regulation of Cx43 with siRNA knockdown techniques decreased mitochondrial Cx43 content and increased oxidative stress [32]. Furthermore, an increase in mitochondrial size was observed in Cx43-knockdown adipocytes compared to controls [32]. Though this study provides insight into the role of Cx43 on mitochondrial function and structure, the results do not make it possible to distinguish between the effects of mitochondrial Cx43 knockdown versus effects of knockdown of Cx43 in other parts of the cell, such as the gap junction plaques on the cell surface. However, the level of mitochondrial Cx43, detected with Western blot and immunocytochemistry, was demonstrated to increase three-fold, following stimulation with a beta3-adrenergic receptor agonist, CL316 243, while the level of Cx43 at the plasma membrane did not change. The authors suggested, since the metabolic activity increased following treatment with the beta3-adrenoceptor agonist in adipocytes with higher levels of mitochondrial Cx43, that mitochondrial Cx43 could serve a functional role in protecting the cell from increased metabolic activity [32,33].

Additional support for the role of mitochondrial Cx43 was provided by innovative studies in which mitochondrial Cx43 was replaced with Cx32, which has lower potassium conductance, and the mitochondrial potassium influx was reduced [99]. Furthermore, the S-nitrosylation of mitochondrial Cx43 under ischemic conditions increased mitochondrial permeability to potassium, consistent with the role of mitochondrial Cx43 in potassium influx. To date, the identity of the mitochondrial Cx43-sensitive potassium channel is ill-defined. However, the ATP-sensitive (mitoKATP), Ca^2+^-activated and voltage-gated Kv1.3 potassium channels, which are all present in the inner mitochondrial membrane, are potential channels that may be regulated via mitochondrial Cx43 [99,100]. Indeed, while a physical interaction between these channels and Cx43 in the inner mitochondrial membrane has not been established, functional crosstalk between mitochondrial Cx43 and the mitoKATP channel has been demonstrated during ischemia preconditioning [100,101]. Nonetheless, based on the available data, mitochondrial Cx43 stimulates mitochondrial potassium influx by apparently forming hemichannels [38]. The location of mitochondrial Cx43 has been suggested to have pathophysiological implications.

### 1.6. Pathophysiological Significance of Mitochondrial Cx43

Studies examining the expression of mitochondrial Cx43 in cardiovascular and metabolic diseases are limited, largely observational and lack definitive information regarding the functional significance of altered Cx43 expression on mitochondrial function. The age-related loss of Cx43 was reported through immunocytochemical and Western blot analysis within the atrial sinoatrial node of guinea pigs [102], atria of rabbits [103] and ventricles of aged mice [104] and rats [105]. However, only one study in mice showed an age-related decline in mitochondrial Cx43 [46].

There is conflicting information about the relationship between the abundance of Cx43 gap junction plaques in hypertensive hearts with hypertrophy. While some studies show reduced Cx43 in the hypertensive heart [106,107] others showed that it was increased [108]. These contradictory results may be due to differences in sampling procedures at variable time points. Based on Western blot analysis and fluorescence microscopy reports, in the spontaneously hypertensive rat [109] and the volume overload pig model [110], gap junctional Cx43 initially increases in response to pressure overload, but as hypertrophy progresses and the heart becomes severely damaged, junctional Cx43 declines. To date, no studies have directly examined whether the expression of mitochondrial Cx43 is altered in the hypertensive heart. Thus, future studies are warranted to address whether the expression of Cx43 is altered in cardiac mitochondria during hypertension. It would also be relevant to determine whether subsarcolemmal mitochondrial populations are the site of differential Cx43 expression during hypertension.

Very few studies have examined mitochondrial Cx43 expression in diabetes and hypercholesterolemia. The Cx43 distribution was demonstrated to change in a diabetic cardiomyopathy murine model, with Cx43 amounts decreasing at the plasma membrane and increasing on the inner mitochondrial membrane. [111]. However, in diabetic retinopathy model, rat retinal endothelial cells grown in high glucose showed reduced mitochondrial Cx43 localization to the inner mitochondrial membrane [41]. This study associated reduced mitochondrial Cx43 with the disruption of the mitochondrial networks. More recently, it was revealed that the high glucose-induced reduction in mitochondrial Cx43 in rat retinal endothelial cells resulted increased apoptosis, thus suggesting a protective role for mitochondrial Cx43 in diabetic retinopathy [112]. In rats fed a 2% cholesterol-rich diet, Cx43 was decreased in the heart, and when subjected to ischemic preconditioning, these hearts were less protected from ischemic damage, thereby implicating Cx43 expression in cardiac protection during ischemia-reperfusion injury [113]. Unfortunately, the mitochondrial localization of Cx43 was not examined in this study. Together, this still suggests the necessity of understanding how Cx43 localization is altered in obesity or metabolic syndrome disorders. Most of our knowledge on the functional role of mitochondrial Cx43 comes from ischemia/reperfusion models. However, direct evidence of a cause-and-effect relationship has not been established. Ischemia/reperfusion injury models demonstrate that mitochondrial Cx43 can undergo phosphorylation [94] and that phosphorylated mitochondrial Cx43 can interact with the potassium channel, inwardly rectifying (Kir6.1) subunit of the K^+^ ATP channel [101]. This interaction underlies cell protection from ischemic injury by a yet unknown mechanism. Thus, future studies are needed to fill this knowledge gap.

Furthermore, while the role of mitochondrial Cx43 in the neural system has not been examined, Cx43 has been reported in the mitochondria of astrocytes isolated from mouse brains. It has been speculated that the mitochondrial Cx43 plays a role in the regulation of mitochondrial K^+^ uptake [40], but the role in mitochondrial function and possible pathophysiology remains unclear and there is a need for further study. This is particularly necessary given that there is loss of Cx43 expression through demyelination and an increase in Cx43 expression in Alzheimer’s disease. Demyelination, however, has been suggested to contribute to Alzheimer’s disease [114]. The measured changes in Cx43 expression are therefore difficult to understand. There is a need to explore beyond only protein expression and to determine how changes in cellular localization, e.g., in mitochondria, are impacted by these disorders.

## 2. Summary

Here, we sought to highlight what is known about mitochondria gap junction interactions, as well as the potential roles of connexins in mitochondria. Mitochondrial association with gap junction-containing structures has been demonstrated across cell types, but the benefit of such contact areas has only been speculated to bring connexins into close contact, so that they can be used by the mitochondria. Experiments are needed to demonstrate how changes in gap junction plaque and annular gap junction vesicle formation affect contact areas between the two structures and mitochondrial connexin levels and mitochondrial functions. Mitochondrial connexin trafficking, location (inner versus outer membrane) and changes in mitochondrial function that are dependent on increased or decreased connexin expression need to be determined. Note, methods that distinguish between changes in connexins in the entire cell versus changes in mitochondrial connexins are also required to further investigate the roles of gap junction-mitochondrial contacts and the significance of these contact sites and mitochondrial connexin abundance to physiological events.

The trafficking of connexin to the mitochondria is thought to involve Hsp90 and TOM 20, but the role, if any, of contact sites in connexin trafficking to the mitochondria and the mechanisms associated with connexin transfer from the outer to the inner mitochondrial membrane has not been studied. Furthermore, the methods for removing old connexins from the inner membrane or their degradation have never, to our knowledge, been studied, despite extensive studies on the degradation of the cell surface-localized connexins.

Mitochondrial connexins are known to be phosphorylated, but the regulatory molecules needed for this phosphorylation are unknown. Much of what is known about mitochondrial connexins is based on indirect methods of analysis. Further, whether mitochondrial connexins are assembled into functional hemichannels once on the inner mitochondrial membrane and the underlying mechanisms of this assembly have not been determined. Short connexin isoforms (20–30 kDa, e.g., GJA1-20k) are synthesized as a result of alternative translation start sites and are localized to the plasma membrane [115]. The GJA1-20k isoform has been demonstrated on the mitochondrial outer membrane, where it is thought to protect mitochondria against oxidative stress [116]. Further studies are required to determine the mechanisms that facilitate the localization of connexin and its isoforms to the mitochondria. A putative working model to explain the relationship between mitochondrial contact sites, Cx43 translocation, the role of mitochondrial Cx43 in potassium flux and pathophysiology is shown in Figure 6. In this model, Cx43 derived from the gap junction plaques, annular gap junction membranes or other sources would be translocated into the outer mitochondria membrane via the HSP90 and TOM 20 translocation complex [70,74]. Interaction with translocase of the inner membrane (TIM) would then result in the translocation of Cx43 to the inner mitochondrial membrane where it remains in monomeric form or is assembled into functional hemichannels by an unknown mechanism [5,74]. The hemichannels, once in the inner membrane, could then participate in different mitochondrial functions. The interaction of the hemichannels with complex 1 of the electron transport chain would increase oxidative phosphorylation and lead to increased ATP production, which in turn could inhibit apoptosis [75,85]. In addition, the interaction of the hemichannels with the K^+^ channel could increase potassium flux [87,88]. Furthermore, hemichannels may directly block reactive oxygen species (ROS) production [5]. The possible combined effect of inhibiting ROS and apoptosis could lead to ischemic preconditioning. Although previous studies provide a basis for this proposed model, future studies are clearly needed to investigate and verify the steps in this working model.

Translationally, the presence of connexin in the mitochondria has been confirmed and the orientation of connexin on the inner mitochondrial membrane has been demonstrated. However, data regarding a relationship between mitochondrial Cx43 and pathophysiology are conflicting and, although it has been suggested that mitochondrial connexins may play a role in cardiovascular and metabolic pathophysiology, no clear conclusion can be drawn regarding the effects of altered mitochondrial connexin abundance on the metabolism in normal and diseased states.

Overall, we have provided an extensive overview of the major gaps in knowledge regarding contacts between mitochondria and gap junction structures and the structure and function of mitochondrial connexins. Despite advancements in this area, this paper also emphasizes the need for further experimental data to prove or disprove current hypotheses regarding mitochondrial-gap junction contacts and mitochondrial connexins and their roles in mitochondrial function, cellular physiology, and pathogenesis. Here, we focused on connexins; however, another family of gap junction proteins, pannexins, has been reported at mitochondrial/ER contact sites [87]. Since connexins share many of the same properties, it is tempting to speculate that pannexins are in the mitochondria. The details of pannexin trafficking and function awaits further study. By bringing the experimental findings and putative hypotheses together in this review, we hope to promote further discussions around mitochondrial gap junction proteins, and we are optimistic about future investigations that will uncover more about mitochondrial connexins and their functional and pathophysiological significance.

## Figures and Tables

**Figure 1 ijms-24-09036-f001:**
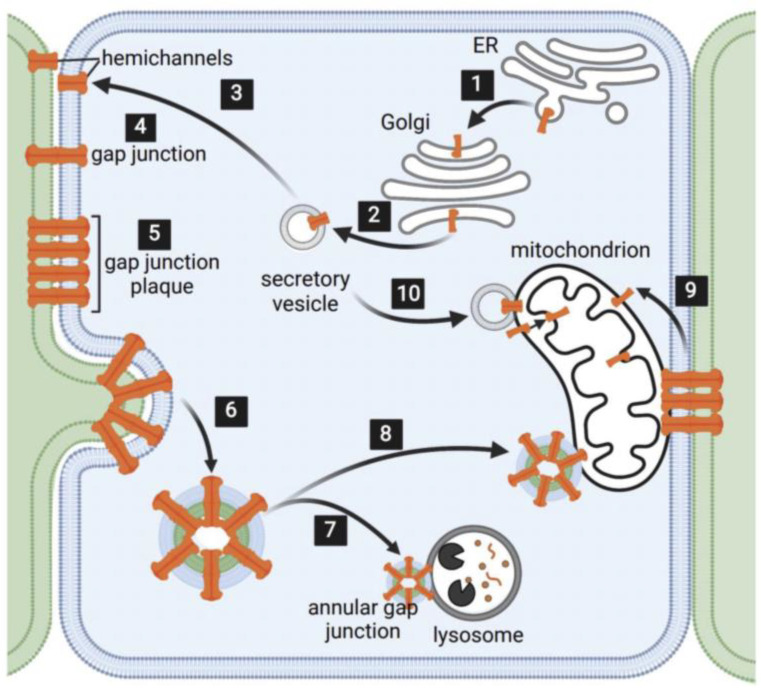
Schematic diagram of connexin trafficking, gap junction plaque formation, internalization and proposed mechanisms for connexin delivery to the mitochondrial membrane. (1) Connexins are synthesized in the ER and (2) oligomerized into hexameric hemichannels in the Golgi. The hemichannels are then transported to plasma membrane in (3) secretory vesicles where they (4) dock with hemichannels on adjacent cells to form a gap junction channels. The channels cluster to form (5) functional gap junction plaques. Gap junction plaques are removed from the plasma membrane by an invagination process that eventually results in the release of (6) annular gap junction vesicles, which contain membranes from both the donor and host cell. (7) Annular gap junctions are (7) degraded by lysosomes. The proposed mechanisms for the delivery of connexins to the mitochondrial include the close physical associations with (8) annular gap junctions, (9) gap junction plaques and (10) secretory vesicles. Figure created with https://www.biorender.com, accessed on 27 April 2023.

**Figure 2 ijms-24-09036-f002:**
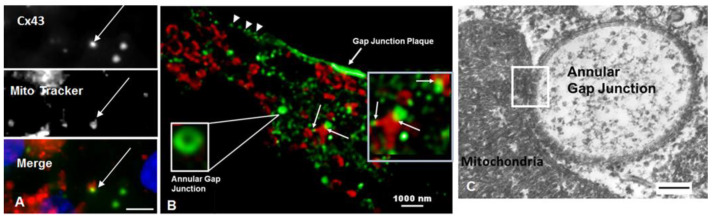
Microscopy images demonstrating gap junction structures in association with mitochondria. (**A**) Immunofluorescent images showing colocalization (yellow in the merged image) of annular gap junctions (arrows) stained with anti-Cx43 antibody (green) and mitochondria stained with Mitotracker (red). (**B**) Superresolution-stimulated emission depletion (STED) microscopy of cells stained with anti-Cx43 antibody (green) and Mitotracker (red), which demonstrates annular gap junctions, cell surface puncta representative of gap junction formation (arrowheads), gap junction plaques and the close associations between mitochondria and annular gap junctions (inset on right (arrows). (**C**) Transmission electron microscopy image revealing the physical contact between an annular gap junction and mitochondria. Scale bars = (**A**): 10 nm, (**B**): 1000 nm, (**C**) 100 nm. Image from reference [6].

**Figure 3 ijms-24-09036-f003:**
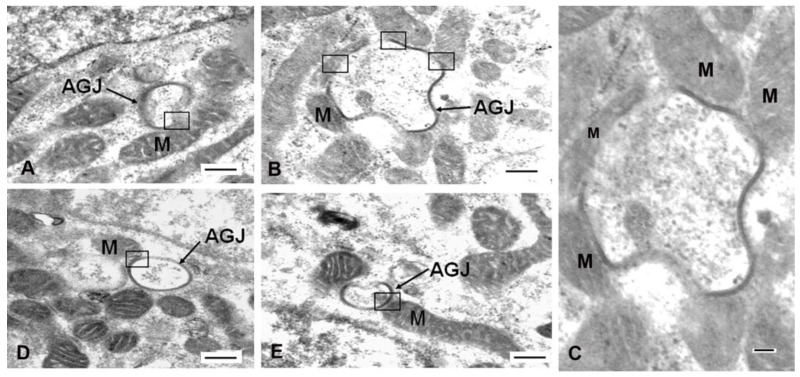
Microscopy images demonstrating gap junction structures in association with mitochondria. (**A**–**E**) Microscopy images demonstrate contact sites between mitochondria (M) and annular gap junctions (AGJ). Black boxes indicate contact between annular gap junctions and mitochondria. (**C**) An enlarged image of the area shown in (**B**) illustrating continuity between annular gap junctions and mitochondria. Scale bars: 500 nm. Image from reference [6].

**Figure 4 ijms-24-09036-f004:**
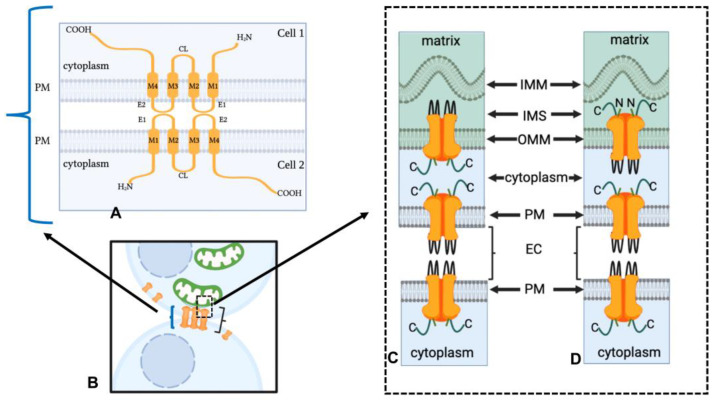
The molecular architecture of Cx43 on the plasma membrane and putative orientation at the mitochondrial/connexin contact site. (**A**) Cx43 has four transmembrane domains (M1–M4). The amino-terminus, carboxyl-terminus and cytoplasmic loop (CL) extend into the cytoplasm, while the two extracellular loops face the extracellular space. In gap junctions, the extracellular loops (E1 and E2) of two apposing cells are in close contact. (**B**) The Cx43 configuration seen in A is depicted in the gap junction plaque shown between two cells (bracket). At the contact site between the mitochondria and gap junction plaque membranes two possible configurations have been suggested (dashed box in (**B**) and the entirety of (**C**,**D**)). Either (**C**) the hemichannels on the outer mitochondrial membranes oriented with the carboxyl terminus facing the cytoplasm and carboxyl terminal to carboxyl docking would be possible or (**D**) hemichannels or connexin monomers on the outer mitochondrial membrane have their extracellular loops oriented toward the cytoplasm, which would allow for interactions between the “extracellular loop” and the carboxyl terminus. IMM = inner mitochondrial membrane; OMM = outer mitochondrial membrane; C = carboxyl terminus; PM = plasma membrane; IMS = Intermembrane space; EC = extracellular space. Figure created with https://www.biorender.com. Given the orientation of the connexin in the plasma membrane with the extracellular loops in the extracellular space, it is unlikely that mitochondrial connexins or other possible proteins found on the outer membrane of the mitochondria could interact with the amino acids of the gap junction plaque or annular vesicle connexin extracellular loops. Instead, it has been suggested that binding may occur between the carboxyl-terminal tail segments of connexins, which extend into the cytoplasm (Figure 4C). Specifically, in this proposed model, binding would occur between the carboxyl-terminal regions of the mitochondria connexin and gap junction plaque or annular vesicle connexin molecules (Figure 4C). Support for binding at the carboxyl-terminal regions is provided by the findings that carboxyl-terminal regions can dimerize [77,78,79]. During the gap junction plaque internalization process, which results in the release of annular gap junction vesicles within the cytoplasm, the docked hemichannels remain bound at their extracellular loops [22,24,26,31,80,81]. Thus, the annular gap junction vesicle has a double membrane that is derived from the membrane of the donor cell and the membrane of the recipient cell, and the extracellular loops are located within the space between these two membranes. The carboxyl-terminal region of the connexin hemichannels is derived from the membrane of the host cell, and thus, they are located within the cytoplasm. The carboxyl-terminal region tail of annular gap junction vesicle channels would therefore potentially be available for direct or indirect interaction with the proteins (e.g., connexins) on the outer mitochondrial membrane. However, interactions between the carboxyl-terminal tail and the “external loops” (E1 and E2) (the regions of the connexin molecule that would be docked in the extracellular space, but depending on the orientation on outer mitochondrial membrane, could be projecting into the cytoplasm) (Figure 4D) cannot be ruled out and have been also suggested [9].

**Figure 5 ijms-24-09036-f005:**
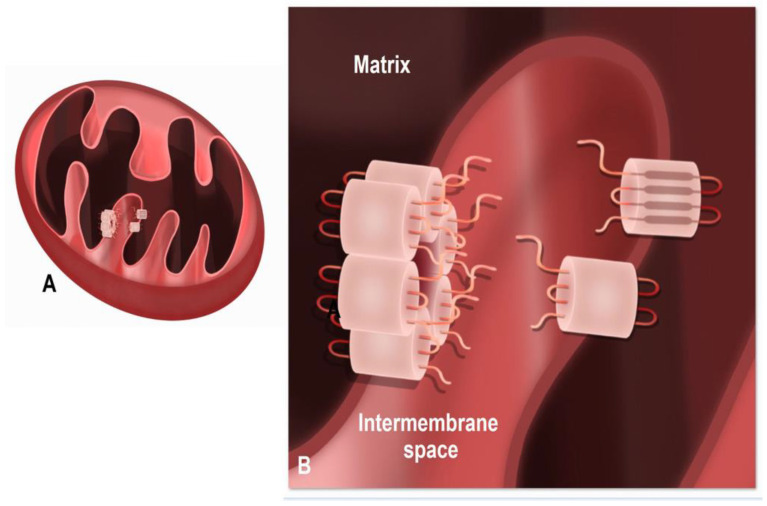
Connexin (Cx43) orientation in the inner mitochondrial membrane. (**A**) Connexins localize to the inner mitochondrial membrane as single channels and/or in hexameric hemichannel structures. (**B**) The enlarged illustration of mitochondrial cristae depicts inner mitochondrial membrane-localized connexin oriented, such that extracellular loops extend into the matrix while carboxyl-terminal and N-terminal regions extend into the intermembrane space. The figure was produced using Procreate software version 5.3.2.

**Figure 6 ijms-24-09036-f006:**
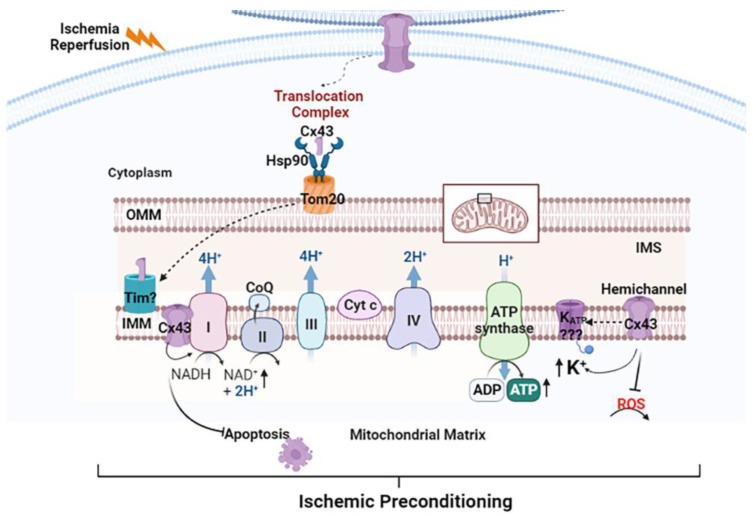
Proposed model for mitochondrial connexin trafficking and possible role in ischemic preconditioning. Connexin trafficking from the gap junction plaques, annular gap junction vesicles or other locations to the inner mitochondrial membrane is facilitated by a Hsp90 and Tom 20 complex and TIM. Monomeric Cx43 or that in hemichannels would then interact with complex I of the electron transport chain to increase oxidative phosphorylation, which in turn inhibits apoptosis. The interaction of hemichannels with the K^+^ channel increases potassium production. Hemichannels can also block ROS production. The reduction in ROS and the blocking of apoptosis lead to ischemic preconditioning. IMM = inner mitochondrial membrane; IMS = intermembrane space; OMM = outer mitochondrial membrane; TIM = translocase of the inner membrane; TOM = translocase of the outer membrane; HSP90 = heat shock protein 90; ROS = reactive oxygen species; Cx43 = Connexin 43. Figure created using https://www.biorender.com.

## Data Availability

No new data were created or analyzed in this study. Data sharing is not applicable to this article.

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
