# Peer review of "Mitochondrial Connexins and Mitochondrial Contact Sites with Gap Junction Structure"

_ijms, 2023, doi:10.3390/ijms24109036_

Round 1
Reviewer 1 Report
Selma Cetin-Ferra et al. review “Mitochondrial Connexins and Contact Sites: An overview.” The review addresses the topic of non-canonical connexin localizations and functions in physiological and disease settings. This review topic is timely. The understanding of the significance of mitochondrial connexins and their connexin contact sites is essential both from a fundamental biology perspective and by opening a window into pathological conditions that my foster the development of therapeutic interventions in diseases linked to mitochondria.
I endorse the review. My comments should be considered as advice for improvements or clarifications in some areas.
The title does not reflect that there are two topics in this review, annular gap junctions and mitochondrial gap junctions. Consider modifying.
The authors write starting in line 100 “However, based on the observations made with a combination of different imaging techniques and the development of new tools for visualizing, localizing, and tracking connexins, it has become evident that contrary to the dogma that connexins only participate in cell-cell communication between cells, they are found on mitochondrial membrane and thus may have other functions in the cell unrelated to cell-cell communication [4].
The authors could point out that this is part of a larger shift in experimental observations of non-canonical functions. For example, Cx hemichannels have also been found in the nuclear membrane and nucleus.
Line 105: Specifically, mitochondria have been demonstrated to form contacts with connexin structures and to be localized to the inner mitochondrial membrane of some mitochondria [32,33] and a role of mitochondrial Cx43 in cardioprotective signaling has been speculated [34-38].
Elaborate on “some” mitochondria. Be more specific.
Line 232: Furthermore, the lumen of large gap junctions has been demonstrated with transmission electron microscopy to contain mitochondria and it is thought that annular gap junctions may facilitate the transfer of mitochondria between cells [57-59].
This is an opening to report that exosomes can be loaded with Cx43 and to put this information into the context of recent developments in the exosome field.
Line 301: An explanation could be that the interactions at mitochondrial/gap junction plaque membrane contact sites are indirect and involve intervening molecules.
This is important. What is the evidence from proteome studies? Have proteins been reported that are part of the Cx interactome that point towards a mitochondrial localization.
Chapter on “Presence of Connexins within Mitochondria”
See my comment above. Have Cx proteins been found in proteome studies that investigated purified organelles?
Chapter on “Pathophysiological Significance of Mitochondrial Cx43.”
What is known about a potential localization of Cx43 or Cx30 in mitochondria of astrocytes.
Line 108: What we know about the function and presence of connexins in the mitochondria is only recently emerging and is poorly understood.
While I agree with the statement, I always struggle with the weak modifier “recent.” Some of the literature goes back two decades. Consider modifying.
Line 146: A close association rather than random association was also suggested based on the presence the bridge-like strands of material in the space between the mitochondria and gap junction plaques [40]. In some cases, mitochondria from two different cells were demonstrated to line up adjacent to either side of the gap junction plaque membrane rather than being randomly distributed near the gap junction plaque membrane [40]. Remarkably, in mice over 40% of the length of gap junction plaque in the ventricular wall was observed to be closely associated with mitochondria [40].
This seems an important observation. Does scientific rigor allow to argue that the proximity reflects that energy producing mitochondria line up with energy demanding processes in the ventricular wall?
Every time I see a review with multiple authors I am cringing. Here we have seven authors from different institutions. While this is perhaps a nice gesture I can’t rationalize the why.
Author Response
Please see attached Word document with our point-by point responses.

Reviewer 2 Report
The present article by Cetin-Ferra is an interesting article where the authors have discussed about the mitochondrial connexins and their significance. This is an excellent piece of work, nicely written but has a lot of typological and spacing errors throughout the script.
Some points to consider
Line 23: Please fix the spacing error.
Line 27: the meaning of sentence is not clear. Please reframe the sentence.
Line 51: Please fix the spacing error.
Line 60; 64, 66, 72, 77, 108, 114: Spacing error.
Check the English of Figure Legend in Fig. 1.
Line 97: The sentence seems to be incomplete. Please check it.
Line 119: Please check the redundancy.
Line 124 to 130: The sentence is too lengthy making its meaning unclear. Please split it in two to three sentences.
Line 162, 164 and 172: Please fix the spacing errors.
Line 270 and 281: Please fix the spacing errors.
Line 362, 408: Spacing error
Line 429 and 430: Spacing error.
English language is fine.
Author Response
Please see the attached Word document.

Reviewer 3 Report
The review of Cetin-Ferra et al. presents the state of the art on mitochondrial connexins and their potential significance for the communication between cellular compartments. As the knowledge on this topic is still fragmentary, the Authors also cite an array of hypotheses to be verified in future. This is a nice “speculative” review that I recommend for publication in IJMS after some minor corrections:
- lines. 60-70: a short note on other determinants of channel permeability (molecular shape and charge) would be add to the message of this fragment;
- l. 75-85: the Authors introduce aGJ here, but short summary on annular gap junctions and their functions in GJ turn-over would clarify the message of this fragment for non-specialists;
- l. 154: the Authors mention that “gap junction dock with hemichannels”. However, they specify this point later in the text: a reference to chapter (3) would clarify this point;
- l. 233: “lumen of gap junctions”; please specify/correct;
At places, the text is difficult to follow due to its complicated syntax: please, check the manuscript for "monster phrases" (like 124-130; 239-245, to mention a few);
Reviewer 4 Report
This review is really enjoyable to read. Authors carefully and critically go through the literature while still keeping it at moderate length.
All parts are built up and connected point by point, so it is convenient for the reader to follow. Also, at each point, readers are reminded to the limitations of the given technique, so that results can be interpreted correctly.
Fig. 1. and Fig.4. clearly introduce and summarize the upcoming text, which further assist the reader to follow the manuscript.
I suggest to accept the paper in the current form.
There are some minor issues - just for further consideration.
A PubMed search on the topic shows, that a recent review also deals with connexins in mitochondria. Authors might consider to include it in the references.
https://www.ncbi.nlm.nih.gov/pmc/articles/PMC9289461/
Recently, it become evident that pannexins also appear at mitochondrial membranes. Is it possible, that some of the observations (e.g. hemichannels in the outer membrane) may be indeed attributed to pannexins?
A few typos to be corrected:
Row 119 should read: "in vertebrate and in invertebrate"
Fig 4 legend should read "between two cells"
Fig 6 legend should read "would then interact"
